# Conditional GANs with Auxiliary Discriminative Classifier

## Abstract

Conditional generative models aim to learn the underlying joint distribution of data and labels, and thus realize conditional generation. Among them, auxiliary classifier generative adversarial networks (AC-GAN) have been widely used, but suffer from the problem of low intra-class diversity on generated samples. In this paper, we point out that the fundamental reason is that the classifier of AC-GAN is generator-agnostic, and therefore cannot provide informative guidance to the generator to approximate the target distribution, resulting in minimization of conditional entropy that decreases the intra-class diversity. Motivated by this observation, we propose a novel conditional GAN with auxiliary *discriminative* classifier (ADC-GAN) to resolve the problem of AC-GAN. Specifically, the proposed auxiliary *discriminative* classifier becomes generator-aware by recognizing the labels of the real data and the generated data *discriminatively*. Our theoretical analysis reveals that the generator can faithfully replicate the target distribution even without the original discriminator, making the proposed ADC-GAN robust to the hyper-parameter and stable during the training process. Extensive experimental results on synthetic and real-world datasets demonstrate the superiority of ADC-GAN on conditional generative modeling compared to competing methods.

## 1 Introduction

Generative adversarial networks (GANs) (Goodfellow et al., 2014) have been gained great progress in learning high-dimensional, complex data distribution such as natural images (Karras et al., 2019; 2020b;a; Brock et al., 2019). Standard GANs consist of a generator network that transfers a latent code sampled from a tractable distribution in the latent space to a data point in the data space and a discriminator network that attempts to distinguish between the real data and the generated one. The generator is trained in an adversarial game against the discriminator such that it can replicate the data distribution at the Nash equilibrium of the game. Remarkably, the training of GANs is notoriously unstable to reach the equilibrium, and thereby the generator is prone to mode collapse (Salimans et al., 2016; Lin et al., 2018; Chen et al., 2019). In addition, practitioners are interested in controlling the properties of the generated samples (Yan et al., 2015; Tan et al., 2020) in practical applications. A key solution to address the above issues is conditioning, leading to conditional GANs (Mirza & Osindero, 2014).

Conditional GANs (cGANs) is a family variant of GANs that leverages the side information from annotated labels of samples to implement and train a conditional generator, and therefore achieve conditional image generation from class-label (Odena et al., 2017; Miyato & Koyama, 2018; Brock et al., 2019) or text (Reed et al., 2016; Xu et al., 2018; Zhu et al., 2019). To implement the conditional generator, the common technique nowadays injects the conditional information via conditional batch normalization (de Vries et al., 2017). To train the conditional generator, a lot of efforts focus on effectively injecting the conditional information into the discriminator or classifier (Odena, 2016; Miyato & Koyama, 2018; Zhou et al., 2018; Kavalerov et al., 2021; Kang & Park, 2020; Zhou et al., 2020). Among them, the auxiliary classifier generative adversarial network (AC-GAN) (Odena et al., 2017) has been widely used due to its simplicity and extensibility. Specifically, AC-GAN utilizes an auxiliary classifier that first attempts to recognize the label of data and then teaches the generator to produce label-consistent (classifiable) data. However, it has been reported that AC-GAN suffers from the low intra-class diversity problem on generated samples, especially on datasets with a large number of classes (Odena et al., 2017; Shu et al., 2017; Gong et al., 2019).

In this paper, we point out that the fundamental reason for the low intra-class diversity problem of AC-GAN is that the classifier is agnostic to the generated data distribution and thus cannot provide informative guidance to the generator in learning the target distribution. Motivated by this observation, we propose a novel conditional GAN with an auxiliary discriminative classifier, namely ADC-GAN, to resolve the problem of AC-GAN by enabling the classifier to be aware of the generated data distribution. To this end, the discriminative classifier is trained to distinguish between the real and generated data while recognizing their labels. The discriminative property enables the classifier to provide the discrepancy between the real and generated data distributions analogy to the discriminator, and the classification property allows it to capture the dependencies between the data and labels. We show in theory that the generator of the proposed ADC-GAN can replicate the joint data and label distribution under the guidance of the discriminative classifier at the optima even without the discriminator, making our method robust to hyper-parameter and stable on training. We also discuss the superiority of ADC-GAN compared to two most related works (TAC-GAN (Gong et al., 2019) and PD-GAN (Miyato & Koyama, 2018)) by analyzing their potential issues and limitations. Experimental results clearly show that the proposed ADC-GAN successfully resolves the problem of AC-GAN by faithfully learning the real joint data and label distribution. The advantages over competing cGANs in experiments conducted on both synthetic and real-world datasets verify the effectiveness of the proposed ADC-GAN in conditional generative modeling.

## 2 PRELIMINARIES AND OUR ANALYSIS

### 2.1 GENERATIVE ADVERSARIAL NETWORKS

Generative adversarial networks (GANs) (Goodfellow et al., 2014) consist of two types of neural networks: the generator $G : \mathcal{Z} \to \mathcal{X}$ that maps a latent code $z \in \mathcal{Z}$ endowed with an easily sampled distribution $P_Z$ to a data point $x \in \mathcal{X}$, and the discriminator $D : \mathcal{X} \to [0, 1]$ that distinguishes between real data that sampled from the real data distribution $P_X$ and fake data that sampled from the generated data distribution $Q_X = G \circ P_Z$ implied by the generator. The goal of the generator is to confuse the discriminator by producing data that is as real as possible. Formally, the objective functions for the discriminator and the generator are defined as follows:

$$\min_G \max_D V(G, D) = \mathbb{E}_{x \sim P_X}[\log D(x)] + \mathbb{E}_{x \sim Q_X}[\log(1 - D(x))]. \qquad (1)$$

Theoretically, the learning of generator under an optimal discriminator can be regarded as minimizing the Jensen-Shannon (JS) divergence between the real data distribution and the generated data distribution, i.e., $\min_G \text{JS}(P_X \| Q_X)$. This would enable the generator to recover the real data distribution at its optima. However, the training of GANs is notoriously unstable, especially when lacking additional supervision such as conditional information. Moreover, the content of the generated images of GANs cannot be specified in advance.

### 2.2 AC-GAN

Learning GANs with conditional information can not only improve the training stability and generation quality of GANs but also achieve conditional generation, which has more practical value than unconditional generation in real-world applications. One of the most representative conditional GANs is AC-GAN (Odena et al., 2017), which utilizes an auxiliary classifier $C : \mathcal{X} \to \mathcal{Y}$ to learn the dependencies between the real data $x \sim P_X$ and the label $y \sim P_Y$ and then enforce the conditional generator $G : \mathcal{Z} \times \mathcal{Y} \to \mathcal{X}$ to synthesize classifiable data as much as possible. The objective functions for the discriminator $D$, the auxiliary classifier $C$, and the generator $G$ of AC-GAN are defined as follows[1]:

$$\max_{D,C} V(G, D) + \lambda \cdot \left( \mathbb{E}_{x,y \sim P_{X,Y}}[\log C(y|x)] \right),$$
$$\min_G V(G, D) - \lambda \cdot \left( \mathbb{E}_{x,y \sim Q_{X,Y}}[\log C(y|x)] \right), \qquad (2)$$

where $\lambda > 0$ is a hyper-parameter, and $Q_{X,Y} = G \circ (P_Z \times P_Y)$ denotes the joint distribution of generated data and labels implied by the generator.

---

[1]We follow the common practice in the literature to adopt the stable version instead of the original one.

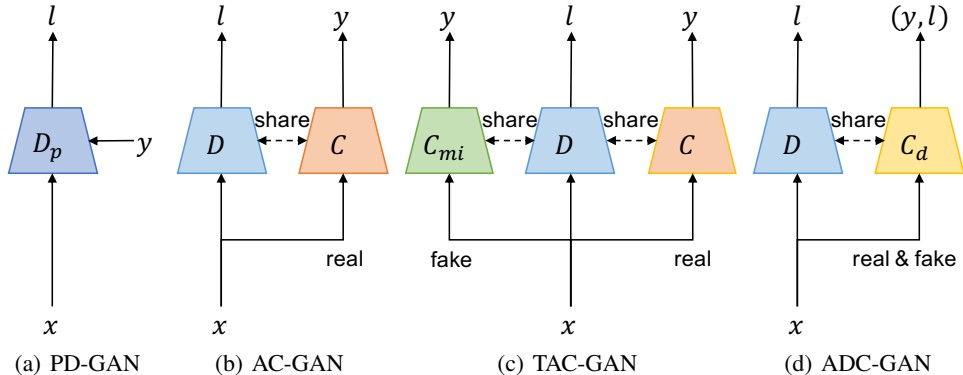

Figure 1: Illustration of discriminators/classifiers of existing conditional GANs (PD-GAN (Miyato & Koyama, 2018), AC-GAN (Odena et al., 2017), and TAC-GAN (Gong et al., 2019)) and the proposed ADC-GAN. $l$ indicates real ($l = 1$) or fake ($l = 0$) and $y$ is the class-label of data $x$. ADC-GAN is different from PD-GAN with explicitly predicting the label and is different with AC-GAN and TAC-GAN that the classifier $C_d$ also distinguishes real from fake like the discriminator.

**Proposition 1.** *The optimal classifier of AC-GAN outputs as follows:*

$$C^*(y|x) = \frac{p(x, y)}{p(x)}. \tag{3}$$

**Theorem 1.** *Given the optimal classifier, at the equilibrium point, optimizing the classification task for the generator of AC-GAN is equivalent to:*

$$\min_G KL(Q_{X,Y} \| P_{X,Y}) - KL(Q_X \| P_X) + H_Q(Y|X), \tag{4}$$

*where $H_Q(Y|X) = -\int \sum_y q(x,y) \log q(y|x) \mathrm{d}x$ is the conditional entropy of generated data.*

The proofs of Proposition 1 and Theorem 1 are referred to Appendix A.1 and A.2, respectively. Our Theorem 1 exposes two shortcomings of AC-GAN. First, maximization of the KL divergence between the marginal generator distribution and the marginal data distribution $\max_G KL(Q_X \| P_X)$ contradicts the goal of conditional generative modeling that matches $Q_{X,Y}$ with $P_{X,Y}$. Although this issue can be mitigated to some extent by the adversarial training objective between the discriminator and the generator that minimizes the JS divergence between the two marginal distributions, we find that it still has a negative impact on the training stability. Second, minimization of the entropy of label conditioned on data with respect to the generated distribution $\min_G H_Q(Y|X)$ will result in that the label of generated data should be completely determined by the data itself. In other words, it will force the generated data of each class away from the classification hyper-plane, explaining the low intra-class diversity of generated samples in AC-GAN especially when the distributions of different classes have non-negligible overlap, which is supported by the fact that state-of-the-art classifiers nor human cannot achieve 100% accuracy on real-world datasets (Russakovsky et al., 2015). Note that the original version of AC-GAN, whose classifier is trained by both real and generated samples, could also suffer from the same issue (see Appendix B).

## 3 THE PROPOSED METHOD: ADC-GAN

The goal of conditional generative modeling is to faithfully approximate the joint distribution of real data and labels regardless of the shape of the target joint distribution (whether there is overlap between distributions of different classes). Note that the learning of the generator in AC-GAN is affected by the classifier. In other words, the reason for the consequence of Theorem 1 originates from Proposition 1, which indicates that the optimal classifier of AC-GAN is agnostic to the density of the generated (marginal or joint) distribution ($q(x)$ or $q(x,y)$). Therefore, the classifier cannot provide the discrepancy between the target distribution and the generated distribution, resulting in a

Table 1: Comparison of objective of the generator under the optimal discriminator and classifier.

| Method | Objective of the generator under the optimal discriminator and classifier |
|---|---|
| AC-GAN | $\min_G \text{JS}(P_X\|Q_X) + \lambda \cdot (\text{KL}(Q_{X,Y}\|P_{X,Y}) - \text{KL}(Q_X\|P_X) + H_Q(Y|X))$ |
| TAC-GAN | $\min_G \text{JS}(P_X\|Q_X) + \lambda \cdot (\text{KL}(Q_{X,Y}\|P_{X,Y}) - \text{KL}(Q_X\|P_X))$ |
| ADC-GAN | $\min_G \text{JS}(P_X\|Q_X) + \lambda \cdot (\text{KL}(Q_{X,Y}\|P_{X,Y}))$ |
| PD-GAN | $\min_G \text{JS}(Q_{X,Y}\|P_{X,Y})$ |

biased learning objective to the generator. Recall that the optimal discriminator $D^*(x) = \frac{p(x)}{p(x)+q(x)}$ is able to be aware of the real data density as well as the generated data density (Goodfellow et al., 2014), and thus can provide the discrepancy $\frac{p(x)}{q(x)} = \frac{D^*(x)}{1-D^*(x)}$ between the real data distribution and the generated data distribution to unbiasedly optimize the generator. Intuitively, the density-aware ability on both real and generated data is caused by the fact that the discriminator attempts to distinguish between real and fake samples. Motivated by this observation, we propose to make the classifier to be distinguishable between real and fake samples, establishing a *discriminative* classifier $C_d : \mathcal{X} \to \mathcal{Y} \times \{0, 1\}$ that recognizes the label of real and fake samples *discriminatively*. Formally, the objective functions for the discriminator $D$, the discriminative classifier $C_d$, and the generator $G$ of the proposed ADC-GAN are defined as follows:

$$\max_{D,C_d} V(G,D) + \lambda \cdot \left( \mathbb{E}_{x,y\sim P_{X,Y}}[\log C_d(y,1|x)] + \mathbb{E}_{x,y\sim Q_{X,Y}}[\log C_d(y,0|x)] \right),$$
$$\min_G V(G,D) - \lambda \cdot \left( \mathbb{E}_{x,y\sim Q_{X,Y}}[\log C_d(y,1|x)] - \mathbb{E}_{x,y\sim Q_{X,Y}}[\log C_d(y,0|x)] \right), \tag{5}$$

where $C_d(y,1|x)$ (reps. $C_d(y,0|x)$) denotes the probability that a data $x$ is classified as the label $y$ and real (reps. fake) data simultaneously.

**Proposition 2.** *For fixed generator, the optimal classifier of ADC-GAN outputs as follows:*

$$C_d^*(y,1|x) = \frac{p(x,y)}{p(x)+q(x)}, C_d^*(y,0|x) = \frac{q(x,y)}{p(x)+q(x)}. \tag{6}$$

The proof is referred to Appendix A.3. Proposition 2 confirms that the discriminative classifier be aware of the densities of the real and generated joint distributions, therefore it is able to provide the discrepancy $\frac{p(x,y)}{q(x,y)} = \frac{C_d^*(y,1|x)}{C_d^*(y,0|x)}$ to unbiasedly optimize the generator as we prove below.

**Theorem 2.** *Given the optimal classifier, at the equilibrium point, optimizing the classification task for the generator of ADC-GAN is equivalent to:*

$$\min_G KL(Q_{X,Y}\|P_{X,Y}). \tag{7}$$

The proof is referred to Appendix A.4. Theorem 2 suggests that the classifier itself can guarantee the generator to replicate the real joint distribution in theory regardless of the shape of the joint distribution. In practice, we retain the discriminator to train the generator and share all layers but the head of the classifier with the discriminator as illustrated in Figure 1 and Equation 5 for faster convergence speed. Coupled with the adversarial training against the discriminator, the generator of the proposed ADC-GAN, under the optimal discriminator and classifier, can be regarded as minimizing the following divergences: $\min_G \text{JS}(P_X\|Q_X) + \lambda \cdot \text{KL}(Q_{X,Y}\|P_{X,Y})$. Since the optimal solution of conditional generative modeling belongs to the optimal solution set of generative modeling, i.e., $\arg\min_G \text{KL}(Q_{X,Y}\|P_{X,Y}) \subseteq \arg\min_G \text{JS}(P_X\|Q_X)$, learning with the discriminator will not change the convergence point of the generator that approximates the joint distribution of real data and labels regardless of the value of hyper-parameter $\lambda > 0$. Furthermore, the hyper-parameter $\lambda$ provides the flexibility to adjust the weight of conditional generative modeling.

## 4 DISCUSSION ON COMPETING METHODS

In this section, we analyze the drawbacks of the two competing methods, TAC-GAN (Gong et al., 2019) and PD-GAN (Miyato & Koyama, 2018), to demonstrate the superiority and rationality of ADC-GAN compared to them. Before diving into the details, we show diagrams of the discriminator and classifier of these methods in Figure 1 and summarize the theoretical learning goal for the generator under the optimal discriminator and classifier of these methods in Table 1 for an overview.

### 4.1 TAC-GAN

TAC-GAN (Gong et al., 2019) addresses the low intra-class diversity issue of AC-GAN by eliminating the conditional entropy with respect to the generated data distribution $H_Q(Y|X)$ via learning of the generator with another classifier $C_{mi} : \mathcal{X} \to \mathcal{Y}$, which is trained on the generated samples. The objective functions for the discriminator $D$, the twin classifiers $C$ and $C_{mi}$, and the generator $G$ of TAC-GAN are defined as follows:

$$\max_{D,C,C_{mi}} V(G,D) + \lambda \cdot \left( \mathbb{E}_{x,y \sim P_{X,Y}}[\log C(y|x)] + \mathbb{E}_{x,y \sim Q_{X,Y}}[\log C_{mi}(y|x)] \right),$$
$$\min_G V(G,D) - \lambda \cdot \left( \mathbb{E}_{x,y \sim Q_{X,Y}}[\log C(y|x)] - \mathbb{E}_{x,y \sim Q_{X,Y}}[\log C_{mi}(y|x)] \right). \tag{8}$$

**Theorem 3.** *Given the twin optimal classifiers, at the equilibrium point, optimizing the classification tasks for the generator of TAC-GAN is equivalent to:*

$$\min_G KL(Q_{X,Y}\|P_{X,Y}) - KL(Q_X\|P_X). \tag{9}$$

The proof is referred to Appendix A.5. Theorem 3 reveals that the learning objective of the generator of TAC-GAN, under the optimal classifier, can be regarded as minimizing contradictory divergences, i.e., minimization between joint distributions but maximization between marginal distributions. Although theoretically the JS divergence or others (Nowozin et al., 2016; Arjovsky et al., 2017) introduced through the adversarial training between the discriminator and the generator might remedy this issue, the optimal discriminator and classifier are difficult to obtain in the practical optimization to ensure that the contradiction is eliminated. We argue that the training instability of TAC-GAN reported in the literature (Kocaoglu et al., 2018; Han et al., 2020) and founded in our experiments can be explained by this analysis and interpretation.

### 4.2 PD-GAN

PD-GAN (Miyato & Koyama, 2018) injects the conditional information into the projection discriminator $D_p : \mathcal{X} \times \mathcal{Y} \to [0,1]$ via the inner-product between the embedding of label and the representation of data to calculate the joint discriminative score of the data-label pair. In such a way, PD-GAN inherits the property of convergence point similar to the standard GAN such that can avoid the low intra-class diversity issue of AC-GAN. The objective functions for the projection discriminator $D_p$ and the generator $G$ of PD-GAN are defined as follows:

$$\min_G \max_{D_p} V(G,D_p) = \mathbb{E}_{x,y \sim P_{X,Y}}[\log D_p(x,y)] + \mathbb{E}_{x,y \sim Q_{X,Y}}[\log(1 - D_p(x,y))]. \tag{10}$$

Based on this minimax game, the optimal projection discriminator has the following form:

$$D_p^*(x,y) = \frac{1}{1 + \exp(-d^*(x,y))} = \frac{p(x,y)}{p(x,y) + q(x,y)}$$
$$\Rightarrow d^*(x,y) = \log \frac{p(x,y)}{q(x,y)} = \log \frac{p(x)}{q(x)} + \log \frac{p(y|x)}{q(y|x)} := r(x) + r(y|x), \tag{11}$$

where $p(y|x) = \frac{\exp(v_y^p \cdot \phi(x))}{\sum_{k=1}^K \exp(v_k^p \cdot \phi(x))}$ and $q(y|x) = \frac{\exp(v_y^q \cdot \phi(x))}{\sum_{k=1}^K \exp(v_k^q \cdot \phi(x))}$ with $K = |\mathcal{Y}|$ is the number of labels. And they accordingly define:

$$r(x) := \psi(\phi(x)),$$

$$r(y|x) := \underbrace{(v_y^p - v_y^q) \cdot \phi(x)}_{\hat{r}(y|x)} - \underbrace{\left( \log \sum_{k=1}^K \exp(v_k^p \cdot \phi(x)) - \log \sum_{k=1}^K \exp(v_k^q \cdot \phi(x)) \right)}_{\text{ⓐ}}. \tag{12}$$

However, PD-GAN actually ignores the partition term ⓐ[2] in Equation 12, and constructs the logit of the projection discriminator in the form of:

$$d(x,y) = r(x) + \hat{r}(y|x) = \psi(\phi(x)) + v_y \cdot \phi(x), \tag{13}$$

---

[2]The authors mistakenly argue that ⓐ can be merged into $r(x)$. However, $r(x)$ does not incorporate any label information ($v^p$ or $v^q$), which should be considered by ⓐ. Therefore, it is unreasonable to merge ⓐ into $r(x)$. PD-GAN actually discards ⓐ in implementing the projection discriminator.

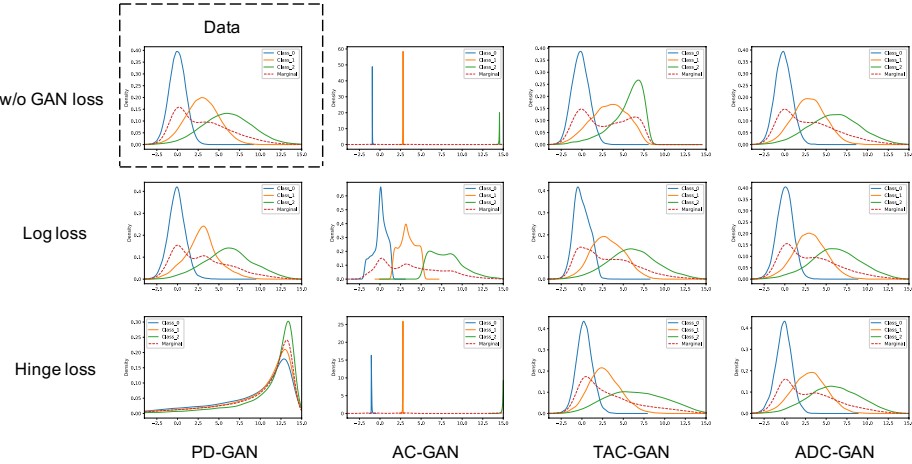

Figure 2: Distribution learning results on one-dimensional synthetic data.

with $v_y = v_y^p - v_y^q$ is the difference between two learnable embeddings of label $y$ defined in two implicit conditional probabilities $p(y|x)$ and $q(y|x)$, $\phi(\cdot)$ is the representation extractor, and $\psi(\cdot)$ outputs a scalar based on the extracted representation. Discarding the partition term would make PD-GAN no longer belong to probability models that model the conditional probabilities $p(y|x)$ and $q(y|x)$, resulting in losing the complete dependencies between data and labels. Moreover, the discriminator constructed according to the optimal form of the minimax GAN lacks theoretical guarantee when applied on other loss functions such as the hinge loss (Lim & Ye, 2017; Tran et al., 2017), which PD-GAN actually used, and may even limit its discriminative ability. The proposed ADC-GAN can be flexibly applied to any version of the loss function $V(G, D)$ as we do not require the specific form of the original discriminator.

## 5 EXPERIMENTS

In this section, we conduct extensive experiments on both synthetic and real-world datasets to demonstrate the effectiveness of the proposed ADC-GAN. Specifically, ADC-GAN has the advantages of being robust to the hyper-parameter $\lambda$, stable during the training process, and capable of accurately modeling dependencies between data and labels, in addition to the comparable or even better performance on conditional generative modeling than existing cGANs.

### 5.1 SYNTHETIC DATA

We first experiment on a one-dimensional synthetic mixture of Gaussian to validate the distribution learning ability of methods. As shown in the left-top of Figure 2, the real data distribution consists of three classes in which there is non-negligible overlap between them. Both generator and discriminator are multi-layer perceptrons with non-linearity of Tanh. In particular, we investigate three different settings on the GAN loss function while keeping the learning of the generator with the classifier fixed if it exists. The first row except the data part shows the learned distributions that are estimated by kernel density estimation (Parzen, 1962) on the generated data of AC-GAN, TAC-GAN, and ADC-GAN without the original GAN loss $V(G, D)$. The second and the third rows show the results of these methods trained with the log loss (Goodfellow et al., 2014) and the hinge loss (Lim & Ye, 2017; Tran et al., 2017), respectively. The poor performance of PD-GAN with hinge loss confirms that it is sensitive to the loss function. AC-GAN tends to generate classifiable data so that it decreases the intra-class diversity in all settings, verifying the Theorem 1. TAC-GAN without GAN loss cannot accurately reproduce the data distribution, verifying the Theorem 3. And the worse performance of TAC-GAN with hinge loss compared with ADC-GAN confirms that the contradiction stated in Theorem 3 is not easy to eliminate by the discriminator. Expectedly, the proposed ADC-GAN can accurately replicate the data distribution even without the original GAN loss, verifying the Theorem 2. For more quantitative results, please refer to Appendix C.1.

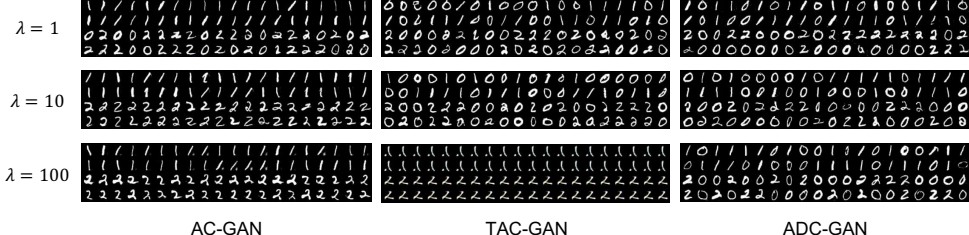

Figure 3: Hyper-parameter robustness results on overlapping MNIST.

## 5.2 OVERLAPPING MNIST

In this subsection, we experiment on a constructed dataset to investigate the hyper-parameter robustness of the proposed ADC-GAN compared to existing classifier-based methods, i.e., AC-GAN and TAC-GAN. We follow the practice of (Gong et al., 2019) to construct a two-class handwritten digit dataset from MNIST. The first class contains an equal number of digits of '0' and '1' and the second class contains an equal number of digits of '0' and '2'. In other words, the support regions of the two classes have an overlap of digits '0'. We change the weight of classifier from $\lambda = 1$ to $\lambda = 100$ with multiplicative step of 10. As shown in Figure 3, AC-GAN mistakenly discards digit '0' in the first class when $\lambda = 1$. When $\lambda \geq 10$, the digit '0' in the generated data of AC-GAN totally disappears, indicating that its classifier encourages the generator to avoid learning the data in the overlapping region. In general, AC-GAN shows a significant decrease in intra-class diversity. TAC-GAN encounters mode collapse when $\lambda = 100$, while ADC-GAN faithfully replicates the real data distribution regardless of the value of $\lambda$. These results suggest that the proposed method has excellent robustness on hyper-parameters compared to existing classifier-based methods since the guidances to the generator received from the discriminator and classifier are harmonious.

## 5.3 CIFAR-10, CIFAR-100, AND TINY-IMAGENET

We experiment on three real-world datasets: CIFAR-10, CIFAR-100, and Tiny-ImageNet. CIFAR-10 consists of 50k training images and 10k validation images with resolution of $32 \times 32$. CIFAR-100 runs similar samples with CIFAR-10 but has 100 classes rather than 10 classes in CIFAR-10. Tiny-ImageNet contains 200 classes where each class contains 500 training images and 50 validation images with resolution of $64 \times 64$. We implement all methods based on the BigGAN-PyTorch repository. The optimizer is Adam with learning rate of $2 \times 10^{-4}$ for both the generator and discriminator on CIFAR-10/100, $1 \times 10^{-4}$ and $4 \times 10^{-4}$ for the generator and discriminator, respectively, on Tiny-ImageNet. We train all methods for 500 epochs with batch size of 50 on CIFAR-10/100 and 100 on Tiny-ImageNet. The discriminator/classifier are updated 4 times per generator update step on CIFAR-10/100, and 2 times on Tiny-ImageNet. We follow the practice of (Miyato & Koyama, 2018; Gong et al., 2019) to adopt the hinge loss (Lim & Ye, 2017; Tran

Table 2: FID and Intra-FID (the averaged intra-class FID) and Accuracy (%) comparison of different methods on CIFAR-10, CIFAR-100, and Tiny-ImageNet, respectively.

| Datasets | Metrics | PD-GAN | AC-GAN | TAC-GAN | ADC-GAN |
|---|---|---|---|---|---|
| CIFAR-10 | FID ($\downarrow$) | 6.51 | 6.81 | 5.85 | **5.56** |
| | Intra-FID ($\downarrow$) | 46.35 | 38.34 | 37.38 | **34.26** |
| | Accuracy ($\uparrow$) | 62.21 | 84.56 | 88.05 | **89.19** |
| CIFAR-100 | FID ($\downarrow$) | 8.47 | 11.63 | 11.37 | **7.98** |
| | Intra-FID ($\downarrow$) | 136.79 | 161.39 | 158.15 | **132.27** |
| | Accuracy ($\uparrow$) | 38.81 | 55.29 | 60.13 | **62.29** |
| Tiny-ImageNet | FID ($\downarrow$) | 20.54 | 24.48 | 23.78 | **20.15** |
| | Intra-FID ($\downarrow$) | **94.23** | 154.82 | 124.68 | 97.69 |
| | Accuracy ($\uparrow$) | 27.56 | 42.26 | 41.78 | **44.29** |

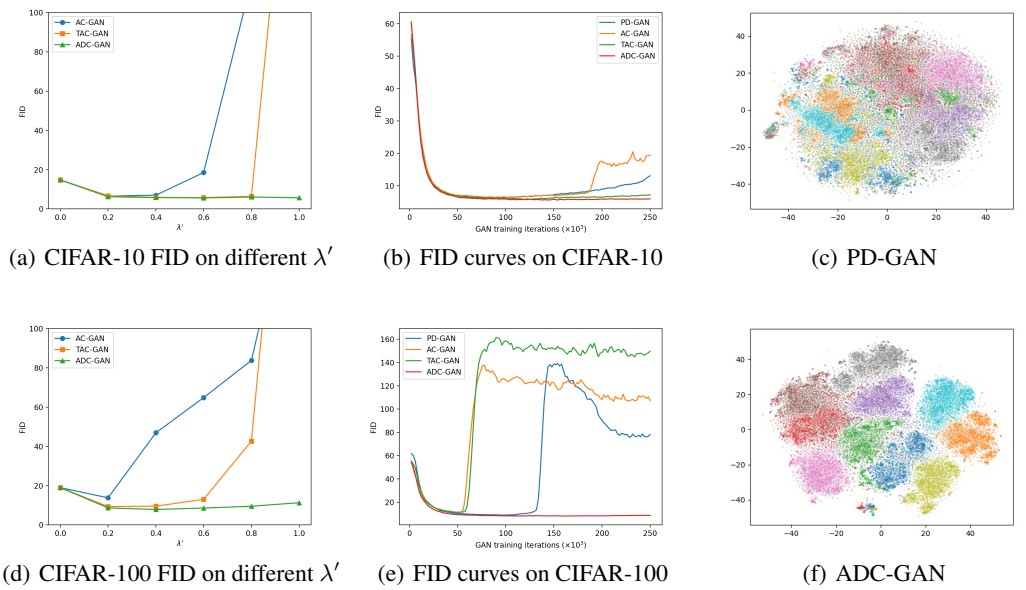

(a) CIFAR-10 FID on different $\lambda'$  (b) FID curves on CIFAR-10  (c) PD-GAN

(d) CIFAR-100 FID on different $\lambda'$  (e) FID curves on CIFAR-100  (f) ADC-GAN

Figure 4: (a,d) FID comparison of methods with different $\lambda'$ on CIFAR-10 and CIFAR-100. The objective of competing methods is $(1-\lambda')V(G,D)+\lambda'V_{\text{C}}(G,C)$, where $V_{\text{C}}(G,C)$ is the task between the generator and classifier. (b,e) FID curves with GAN training iterations on CIFAR-10 and CIFAR-100. (c,f) T-SNE visualization of CIFAR-10 training data, using learned representations extracted from the penultimate layer in discriminators. Different colors represent different classes.

et al., 2017) as an implementation of $V(G,D)$. We set the hyper-parameter of AC-GAN as $\lambda=0.2$ on all datasets as it performs the best. As for TAC-GAN and ADC-GAN, the hyper-parameter is set as $\lambda=1.0$ on CIFAR-10/100 and $\lambda=0.5$ on Tiny-ImageNet.

We report the FID (Heusel et al., 2017) and Intra-FID (Miyato & Koyama, 2018) (the FID for each class) results of all methods on CIFAR-10, CIFAR-100, and Tiny-ImageNet in Table 2. AC-GAN obtains the worst results and diverges on all datasets (see Figure 4(b),4(e),7). TAC-GAN also diverges on CIFAR-100 and Tiny-ImageNet and only achieves a relatively stable FID training curve on CIFAR-10. We here report their results at the best FID checkpoint. These unstable FID curves implicitly reveal the drawback of the existing classifier-based cGANs that minimizes contradictory divergences. To explicitly show this, we set the objective function of classifier-based methods as $(1-\lambda')V(G,D)+\lambda'V_{\text{C}}(G,C)$, where $V_{\text{C}}(G,C)$ is the task between the generator and classifier. As shown in Figure 4(a),4(d), ADC-GAN gains consistent FID results across different $\lambda'$ even for $\lambda'=1.0$ (i.e., without the discriminator), showing strong robustness on $\lambda'$, while AC-GAN and TAC-GAN performs substantially bad when $\lambda'$ becomes large. Table 2 also shows that ADC-GAN achieves the superior FID and Intra-FID on CIFAR-10 and CIFAR-100 and comparable scores with PD-GAN on Tiny-ImageNet. The reason why PD-GAN obtains slightly better Intra-FID than ADC-GAN on Tiny-ImageNet is that Tiny-ImageNet is a more coarse-grained dataset, in which data in different classes have weak correlations. Impressively, ADC-GAN performs much better training stability than PD-GAN in terms of the FID curves as shown in Figure 4(b),4(e),7. We argue that the reason is that ADC-GAN is less prone to over-fitting than PD-GAN due to that the discriminative classifier of ADC-GAN solves a more difficult task than the projection discriminator of PD-GAN.

To investigate whether the model captures accurate dependencies between data and labels, we conduct image classification experiments based on the learned representations extracted from the shared penultimate layer in the discriminator/classifier. Specifically, we utilize the logistical regression classifier in scikit-learn library with default settings to compute the accuracy results. As reported in Table 2, ADC-GAN significantly outperforms competing methods on all datasets, indicating that ADC-GAN effectively learns accurate dependencies between data and labels. The reason is that the discriminative classifier needs to distinguish between real and fake data while simultaneously recog-

Table 3: FID and IS comparison with competing methods on ImageNet. The results of competing methods are copied from TAC-GAN (Gong et al., 2019) and SAGAN (Zhang et al., 2019).

| Datasets | Metrics | PD-GAN/BigGAN | AC-GAN | TAC-GAN | ADC-GAN |
|----------|---------|---------------|--------|---------|---------|
| ImageNet | FID ($\downarrow$) | 22.77 | — | 23.75 | **16.75** |
|          | IS ($\uparrow$) | $38.05 \pm 0.79$ | 28.5 | $28.86 \pm 0.29$ | $\mathbf{55.43 \pm 0.90}$ |

nizing the labels of samples, which forces the classifier to have a more powerful and robust ability of modeling data-to-class relations. Notice that PD-GAN obtains the worst accuracy results. By comparing the CIFAR-10 T-SNE (Van der Maaten & Hinton, 2008) visualization results of PD-GAN and ADC-GAN using the learned features of validation data in Figure 4(c),4(f), one can clearly see that PD-GAN is incapable of learning accurate dependencies between data and labels.

## 5.4 IMAGENET

In this subsection, we compare the proposed ADC-GAN with competing methods on ImageNet ($128 \times 128$) with 1,000 classes (Deng et al., 2009), each of which contains around 1,300 images. We adopt the BigGAN with base channels of $64$ as the backbone of ADC-GAN and train one step for the discriminator/classifier and one step for the generator, following the practices of TAC-GAN (Gong et al., 2019). We follow the instruction of FQ-GAN (Zhao et al., 2020) to train ADC-GAN for 128k iterations. As shown in Table 3, the proposed ADC-GAN significantly outperforms competing cGANs in terms of both Inception Score (IS) (Salimans et al., 2016) and FID metrics, showing the effectiveness on large-scale high-resolution image datasets.

## 6 RELATED WORK

Conditional generative adversarial networks (cGANs) (Mirza & Osindero, 2014) is a family of GANs, which is capable of generating novel data depended on the given label. The research on cGANs can be divided into two aspects. One is to study how to implement a conditional generator network structurally. Approaches in this category are mainly concatenating the label vector with the noise vector (Mirza & Osindero, 2014), conditional batch normalization (de Vries et al., 2017), and conditional convolution layers (Sagong et al., 2019). The other is to study how to train the conditional generator to generate label-dependent data. AC-GAN (Odena et al., 2017) leveraged an auxiliary classifier to determine the relationship between data and labels. MH-GAN (Kavalerov et al., 2021) improved AC-GAN by replacing the cross-entropy loss of the classifier with the multi-hinge loss. Shu et al. (2017) analyzed that AC-GAN learns a biased distribution from a Lagrange view. AM-GAN (Zhou et al., 2018) extended the real class of the discriminator into $K$ classes of the classifier, forming a discriminator with $K+1$ classes. Omni-GAN (Zhou et al., 2020) combined the discriminator with the classifier into a $K + 2$ dimensional multi-label discriminator. TAC-GAN (Gong et al., 2019) corrected the biased learning objective of AC-GAN by introducing another classifier, which was the multi-class version of Anti-Labeler of CausalGAN (Kocaoglu et al., 2018). UAC-GAN (Han et al., 2020) improved the training stability of TAC-GAN via MINE (Belghazi et al., 2018). PD-GAN (Miyato & Koyama, 2018) injected the label information into the discriminator via projection technique. In complementary to our work, ContraGAN (Kang & Park, 2020) modeled the data-to-data relations as well as the data-to-class relations using a conditional contrastive loss.

## 7 CONCLUSIONS

In this paper, we present a novel conditional generative adversarial network with an auxiliary discriminative classifier (ADC-GAN) to achieve faithful conditional generative modeling. The discriminative classifier can provide the discrepancy between the joint distribution of the real data and labels and that of the generated data and labels to the generator by discriminatively predicting the label of the real and generated data. Therefore, the generator can faithfully learn the real joint data and label distribution at the Nash equilibrium. We also discuss the differences between ADC-GAN with competing cGANs and analyze their potential issues and limitations. Extensive experimental results validate the theoretical superiority of the proposed ADC-GAN compared to competing cGANs.

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

# A  PROOFS

## A.1  PROOF OF PROPOSITION 1

**Proposition 1.** *The optimal classifier of AC-GAN outputs as follows:*

$$C^*(y|x) = \frac{p(x,y)}{p(x)}. \tag{3}$$

*Proof.*

$$\max_C \mathbb{E}_{x,y\sim P_{X,Y}}[\log C(y|x)] = \mathbb{E}_{x\sim P_X}\mathbb{E}_{y\sim P_{Y|X}}[\log C(y|x)]$$

$$\Rightarrow \min_C \mathbb{E}_{x\sim P_X}\mathbb{E}_{y\sim P_{Y|X}}[-\log C(y|x)] = \mathbb{E}_{x\sim P_X}[H(p(y|x)) + \mathrm{KL}(p(y|x)\|C(y|x))]$$

$$\Rightarrow C^*(y|x) = \arg\min_C \mathrm{KL}(p(y|x)\|C(y|x)) = p(y|x) = \frac{p(x,y)}{p(x)}$$

$\square$

## A.2  PROOF OF THEOREM 1

**Theorem 1.** *Given the optimal classifier, at the equilibrium point, optimizing the classification task for the generator of AC-GAN is equivalent to:*

$$\min_G KL(Q_{X,Y}\|P_{X,Y}) - KL(Q_X\|P_X) + H_Q(Y|X), \tag{4}$$

*where $H_Q(Y|X) = -\int \sum_y q(x,y)\log q(y|x)\mathrm{d}x$ is the conditional entropy of generated data.*

*Proof.*

$$\max_G \mathbb{E}_{x,y\sim Q_{X,Y}}\left[\log C^*(y|x)\right] = \mathbb{E}_{x,y\sim Q_{X,Y}}\left[\log\frac{p(x,y)}{p(x)}\right] = \mathbb{E}_{x,y\sim Q_{X,Y}}\left[\log\frac{p(x,y)}{q(x,y)}\frac{q(x)}{p(x)}\frac{q(x,y)}{q(x)}\right]$$

$$= \mathbb{E}_{x,y\sim Q_{X,Y}}\left[\log\frac{p(x,y)}{q(x,y)}\right] + \mathbb{E}_{x\sim Q_X}\left[\log\frac{q(x)}{p(x)}\right] + \mathbb{E}_{x,y\sim Q_{X,Y}}\left[\log\frac{q(x,y)}{q(x)}\right]$$

$$\Rightarrow \min_G \mathrm{KL}(Q_{X,Y}\|P_{X,Y}) - \mathrm{KL}(Q_X\|P_X) + H_Q(Y|X)$$

$\square$

## A.3  PROOF OF PROPOSITION 2

**Proposition 2.** *For fixed generator, the optimal classifier of ADC-GAN outputs as follows:*

$$C_d^*(y,1|x) = \frac{p(x,y)}{p(x)+q(x)}, C_d^*(y,0|x) = \frac{q(x,y)}{p(x)+q(x)}. \tag{6}$$

*Proof.*

$$\max_C \mathbb{E}_{x,y\sim P_{X,Y}}[\log C(y,1|x)] + \mathbb{E}_{x,y\sim Q_{X,Y}}[\log C(y,0|x)] \Rightarrow \max_C \mathbb{E}_{x,y,l\sim P_{X,Y,L}^m}[\log C(y,l|x)],$$

with $l \in \{0,1\}$ and $p^m(x,y,l) = p^m(x,y,1) + p^m(x,y,0) = \frac{1}{2}p(x,y) + \frac{1}{2}q(x,y)$.

$$\Rightarrow \max_C \mathbb{E}_{x\sim P_X^m}\mathbb{E}_{y,l\sim P_{Y,L|X}^m}[\log C(y,l|x)] \Rightarrow \min_C \mathbb{E}_{x\sim P_X^m}\mathbb{E}_{y,l\sim P_{Y,L|X}^m}[-\log C(y,l|x)]$$

$$\Rightarrow \min_C \mathbb{E}_{x\sim P_X^m}[H(p^m(y,l|x)) + \mathrm{KL}(p^m(y,l|x)\|C(y,l|x))]$$

$$\Rightarrow C^*(y,l|x) = \arg\min_C \mathrm{KL}(p^m(y,l|x)\|C(y,l|x)) = p^m(y,l|x) = \frac{p^m(x,y,l)}{p^m(x)}$$

Therefore, the optimal classifier of ADC-GAN has the form of $C^*(y,1|x) = \frac{p^m(x,y,1)}{p^m(x)} = \frac{p(x,y)}{p(x)+q(x)}$ and $C^*(y,0|x) = \frac{p^m(x,y,0)}{p^m(x)} = \frac{q(x,y)}{p(x)+q(x)}$ that concludes the proof.

$\square$

### A.4 PROOF OF THEOREM 2

**Theorem 2.** *Given the optimal classifier, at the equilibrium point, optimizing the classification task for the generator of ADC-GAN is equivalent to:*

$$\min_G KL(Q_{X,Y} \| P_{X,Y}). \tag{7}$$

*Proof.*

$$\max_G \mathbb{E}_{x,y \sim Q_{X,Y}} \left[ \log C^*(y,1|x) \right] - \mathbb{E}_{x,y \sim Q_{X,Y}} \left[ \log C^*(y,0|x) \right]$$

$$\Rightarrow \min_G -\mathbb{E}_{x,y \sim Q_{X,Y}} \left[ \log C^*(y,1|x) \right] + \mathbb{E}_{x,y \sim Q_{X,Y}} \left[ \log C^*(y,0|x) \right]$$

$$\Rightarrow \min_G -\mathbb{E}_{x,y \sim Q_{X,Y}} \left[ \log \frac{p(x,y)}{p(x)+q(x)} \right] + \mathbb{E}_{x,y \sim Q_{X,Y}} \left[ \log \frac{q(x,y)}{p(x)+q(x)} \right]$$

$$\Rightarrow \min_G \mathbb{E}_{x,y \sim Q_{X,Y}} \left[ \log \frac{q(x,y)}{p(x,y)} \right] \Rightarrow \min_G KL(Q_{X,Y} \| P_{X,Y})$$

$\square$

### A.5 PROOF OF THEOREM 3

**Proposition 3.** *For fixed generator, the twin optimal classifiers of TAC-GAN output as follows:*

$$C^*(y|x) = \frac{p(x,y)}{p(x)}, C_{mi}^*(y|x) = \frac{q(x,y)}{q(x)}. \tag{14}$$

*Proof.* The proof is similar to that of Proposition 1 in Appendix A.1 by considering $C$ and $C_{mi}$ as two independent classifiers with respect to distribution $P$ and $Q$, respectively. $\square$

**Theorem 3.** *Given the twin optimal classifiers, at the equilibrium point, optimizing the classification tasks for the generator of TAC-GAN is equivalent to:*

$$\min_G KL(Q_{X,Y} \| P_{X,Y}) - KL(Q_X \| P_X). \tag{9}$$

*Proof.*

$$\max_G \mathbb{E}_{x,y \sim Q_{X,Y}} \left[ \log C^*(y|x) \right] - \mathbb{E}_{x,y \sim Q_{X,Y}} \left[ \log C_{mi}^*(y|x) \right]$$

$$\Rightarrow \max_G \mathbb{E}_{x,y \sim Q_{X,Y}} \left[ \log \frac{p(x,y)}{p(x)} \right] - \mathbb{E}_{x,y \sim Q_{X,Y}} \left[ \log \frac{q(x,y)}{q(x)} \right]$$

$$\Rightarrow \max_G \mathbb{E}_{x,y \sim Q_{X,Y}} \left[ \log \frac{p(x,y)}{q(x,y)} \right] - \mathbb{E}_{x \sim Q_X} \left[ \log \frac{p(x)}{q(x)} \right]$$

$$\Rightarrow \min_G KL(Q_{X,Y} \| P_{X,Y}) - KL(Q_X \| P_X)$$

$\square$

## B ISSUE OF THE ORIGINAL AC-GAN

In this section, we show that original AC-GAN whose auxiliary classifier is trained with both real and fake samples still suffer from the issue proved in Theorem 1. Formally, the full objective function of the original AC-GAN is formulated as follows.

$$\max_{D,C} V(G,D) + \lambda \cdot \left( \mathbb{E}_{x,y \sim P_{X,Y}} [\log C(y|x)] + \mathbb{E}_{x,y \sim Q_{X,Y}} [\log C(y|x)] \right),$$
$$\min_G V(G,D) - \lambda \cdot \left( \mathbb{E}_{x,y \sim Q_{X,Y}} [\log C(y|x)] \right). \tag{15}$$

The objective function for training the classifier can be rewritten as:

$$\max_C \mathbb{E}_{x,y \sim P_{X,Y}}[\log C(y|x)] + \mathbb{E}_{x,y \sim Q_{X,Y}}[\log C(y|x)] \Rightarrow \max_C \mathbb{E}_{x,y \sim P^m_{X,Y}}[\log C(y|x)], \quad (16)$$

with $p^m(x,y) = \frac{1}{2}(p(x,y) + q(x,y))$ and $p^m(x) = \sum_y p^m(x,y) = \frac{1}{2}(p(x) + q(x))$. And we can obtain the optimal classifier by the following:

$$
\begin{aligned}
\max_C \mathbb{E}_{x,y \sim P^m_{X,Y}}[\log C(y|x)] &\Rightarrow \min_C \mathbb{E}_{x \sim P^m_X, y \sim P^m_{Y|X}}[-\log C(y|x)] \\
&\Rightarrow \min_C \mathbb{E}_{x \sim P^m_X}[H(p^m(y|x)) + \text{KL}(p^m(y|x)\|C(y|x))] \\
&\Rightarrow C^*(y|x) = p^m(y|x) = \frac{p(x,y) + q(x,y)}{p(x) + q(x)}
\end{aligned}
\tag{17}
$$

Even though the conditional generator learns the joint real data and label distribution, i.e., $q(x,y) = p(x,y)$ and $q(x) = p(x)$, the optimal classifier $C^*(y|x) = \frac{p(x,y)+q(x,y)}{p(x)+q(x)} = \frac{p(x,y)}{p(x)}$ will still provide the objective stated in Theorem 1 to optimize the generator, which contains the conditional entropy of generated samples $H_Q(Y|X)$ that would reduce the intra-class diversity of generated samples. In other words, the discriminative classifier does not allow the generator to remain on the desired distribution because it still provide momentum to update the generator, resulting in a biased learning objective for the generator.

# C  MORE RESULTS

## C.1  SYNTHETIC DATA

In this section, we report more results on experiments conducted on the one-dimensional synthetic data and a new two-dimensional synthetic data. The one-dimensional data consists of three Gaussian components with $\mu_0 = 0, \mu_1 = 3, \mu_2 = 6$ and $\sigma_0 = 1, \sigma_1 = 2, \sigma_2 = 3$, and the similar for the two-dimensional data. For implementing the generator, discriminator, and classifier, we use three-layer multi-layer perceptron with hidden size of 10 and the Tanh non-linearity. The optimizer is Adam with learning rate $\alpha = 0.002$ and betas $(\beta_1, \beta_2) = (0.5, 0.999)$. We train all methods for 40 epochs with batch size of 256. Table 4 reports the quantitative maximum mean discrepancy (MMD) results on the one-dimensional synthetic data conducted in Section 5.1. Lower MMD means better learning results. Figure 6 and Table 5 show the qualitative and quantitative results, respectively, conducted on the two-dimensional synthetic Gaussian data. In general, the proposed ADC-GAN consistently replicates the data distribution under different loss function settings.

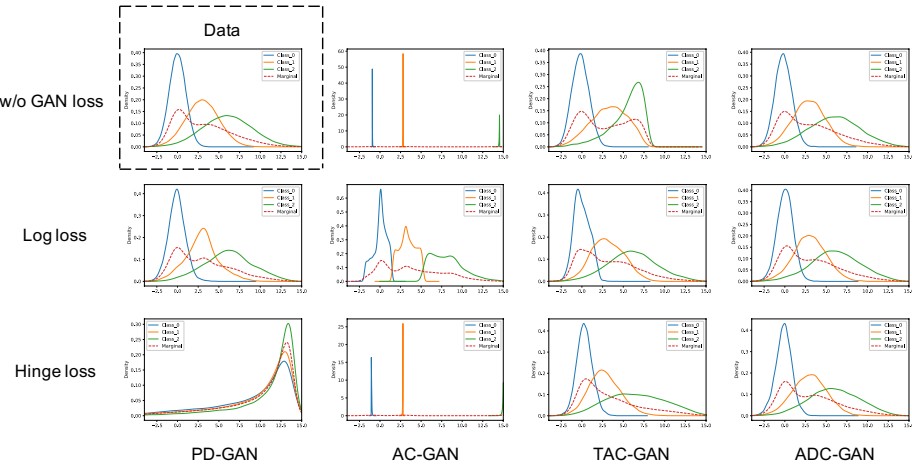

Figure 5: Distribution learning results on the one-dimensional synthetic data.

Table 4: MMD ($\downarrow$) results of each method on the one-dimensional synthetic data.

| GAN Loss | Class | PD-GAN | AC-GAN | TAC-GAN | ADC-GAN |
|---|---|---|---|---|---|
| No | Class0 | - | 1.83 | **0.05** | 0.06 |
|  | Class1 | - | 27.32 | 5.99 | **0.06** |
|  | Class2 | - | 27583.31 | 130.89 | **0.19** |
|  | Marginal | - | 2919.31 | 7.81 | **0.14** |
| Log | Class0 | 0.02 | 0.13 | **0.01** | 0.06 |
|  | Class1 | **0.05** | 0.64 | 0.14 | 0.07 |
|  | Class2 | **0.10** | 639.55 | 0.90 | 0.19 |
|  | Marginal | **0.11** | 75.58 | 0.66 | 0.14 |
| Hinge | Class0 | 11328.12 | 2.08 | **0.21** | 0.40 |
|  | Class1 | 10671.01 | 29.96 | 3.04 | **1.22** |
|  | Class2 | 8420.67 | 32011.93 | 93.08 | **8.55** |
|  | Marginal | 10218.60 | 3243.28 | 7.81 | **0.80** |

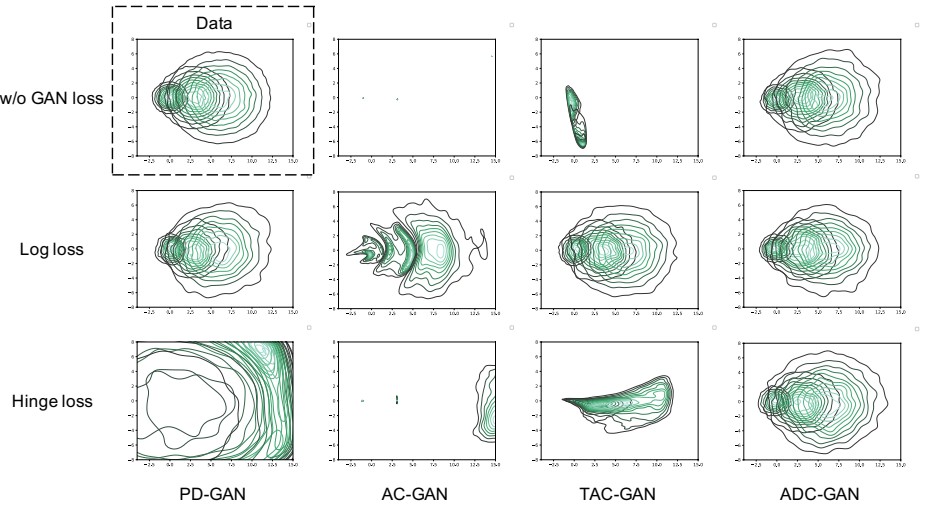

Figure 6: Distribution learning results on the two-dimensional synthetic data.

Table 5: MMD (↓) results of each method on the two-dimensional synthetic data.

| GAN Loss | Class | PD-GAN | AC-GAN | TAC-GAN | ADC-GAN |
|---|---|---|---|---|---|
| No | Class0 | - | 1.28 | 1.55 | **0.07** |
| | Class1 | - | 6.64 | 152.49 | **0.09** |
| | Class2 | - | 10491.93 | 671.89 | **0.11** |
| | Marginal | - | 1085.61 | 166.03 | **0.02** |
| Log | Class0 | 0.05 | 0.06 | **0.01** | 0.02 |
| | Class1 | **0.02** | 0.64 | 0.27 | 0.12 |
| | Class2 | 0.73 | 119.91 | **0.52** | 0.86 |
| | Marginal | **0.01** | 13.12 | 0.04 | 0.08 |
| Hinge | Class0 | 1625.92 | 1.42 | 0.70 | **0.06** |
| | Class1 | 1138.02 | 1.43 | 5.75 | **0.23** |
| | Class2 | 918.97 | 9440.54 | 33.34 | **0.06** |
| | Marginal | 1203.51 | 1019.39 | 7.99 | **0.03** |

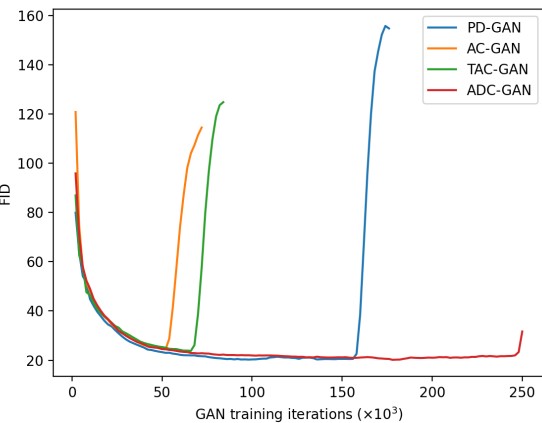

Figure 7: FID curves with GAN training iterations on Tiny-ImageNet.

