# OpenReview forum: "Conditional GANs with Auxiliary Discriminative Classifier"
_ICLR.cc/2022/Conference — ICLR 2022 Submitted_

### Official Review · Reviewer_uPwH · 2021-11-01

**Correctness:** 3
**Technical Novelty And Significance:** 4
**Empirical Novelty And Significance:** 3
**Recommendation:** 8
**Confidence:** 4

**Main Review:**

Strengths:

- The problem of low intra-class diversity of classifier-based cGANs is important.
    - Reasons: The projection-based cGAN, PD-GAN, does not suffer from the low intra-class diversity problem, but it converges more slowly than classifier-based cGANs (see Omni-GAN, arXiv:2011.13074).  The classifier-based cGANs converge faster but suffer from low intra-class diversity. Therefore, it is of great value to improve classifier-based cGANs so that we can completely abandon the PD-GAN converging slowly in practice.
- This paper provides a theoretical perspective for analyzing the loss function of different cGANs.
- The proposed ADC-GAN is very simple to implement without additional computational overhead.

Weaknesses:

- Please detail in the paper how the FID in Table 2 is calculated (for example, how many generated images are used, whether the training set or the validation set is used, and whether the inception model is from PyTorch or tensorflow). In addition, I also recommend including IS in Table 2.
- In Equ. 5, the notations of $C_d(y,1|x)$ and $C_d(y,0|x)$ are a bit confusing. After checking the code in the supplementary material, I understood the meaning of the equation. In fact, $C_d(y,1|x)$ and $C_d(y,0|x)$ are implemented using a fully connected layer with output dimensions of num_classes * 2. I suggest that the author use much clearer notation to make it easier to understand. The author can refer to the notation of equation 8, which is clear.
- In Figure 4e, I am surprised that as a classifier-based cGAN, ADC-GAN did not suffer from mode collapse. The author also does not seem to apply weight decay for the discriminator, as Omni-GAN does. I am not sure if the author has used other regularization techniques to stabilize the training. As far as I know, if do not add regularizations such as weight decay, other classifier-based cGANs will collapse earlier, such as AC-GAN, Multi-hinge GAN, and Omni-GAN. It would be better if the author could explain this phenomenon.
- I know that there is an improved version of AC-GAN (ImAC-GAN), as discussed in section 3.3 of the Omni-GAN paper. ImAC-GAN has a clear performance gain compared to AC-GAN. It would be better if the author could discuss the relationship between ImAC-GAN and ADC-GAN.
- In Figure 4 (c) and (f), the results of T-SNE visualization are not very convincing. In my opinion, the author should not use the discriminator to extract feature representations, because the PD-GAN discriminator is less supervised than ADC-GAN. I suggest that the author use a pre-trained classification model to extract features for a fair comparison.
- In the caption of Figure 4, the author says that the T-SNE uses training data, but in the last paragraph of section 5.3, the author says that the T-SNE uses the validation data.


**Summary Of The Paper:**

- This paper aims to solve the low intra-class diversity on generated images of AC-GAN, a classifier-based cGAN.
- As far as I know, this is an important issue that limits classifier-based cGANs (the counterpart is the projection-based cGAN, i.e, PD-GAN).
- The authors point out that the reason is that the classifier of AC-GAN is generator-agnostic and minimization of conditional entropy decreases the intra-class diversity.
- The authors propose ADC-GAN (auxiliary discriminative classifier) to solve this problem, and theoretical analysis is also presented.


**Summary Of The Review:**

The author proposes a simple but seemingly promising classifier-based cGAN. I hope the author can answer my questions above in detail. I will improve the score based on the author's answer.

---

> ### Author Response · Authors · 2021-11-21
> **Response to Reviewer uPwH**
>
> Thank you very much for recognizing our contribution and thoughtful review.
>
> **Q**1: Please detail in the paper how the FID in Table 2 is calculated (for example, how many generated images are used, whether the training set or the validation set is used, and whether the inception model is from PyTorch or tensorflow). In addition, I also recommend including IS in Table 2.
>
> **R1**: We follow the practice of the BigGAN-PyTorch repository to calculate FID in Table 2 with 50k generated images. The reference is the training set, and the inception model is from PyTorch. In this work, our goal is to resolve the low intra-class diversity of AC-GAN. As IS is not capable of measuring the intra-class diversity, we do not report them in Table 2. Instead, we use intra-FID which can measure the intra-class diversity to compare methods.
>
> **Q2**: In Equ. 5, the notations of Cd(y,1|x) and Cd(y,0|x) are a bit confusing. I suggest that the author use much clearer notation to make it easier to understand.
>
> **R2**: We would like to thank you for the valuable suggestion. We will give a detailed description in the updated version.
>
> **Q3**: In Figure 4e, I am surprised that as a classifier-based cGAN, ADC-GAN did not suffer from mode collapse. I am not sure if the author has used other regularization techniques to stabilize the training. It would be better if the author could explain this phenomenon.
>
> **R3**: We implement ADC-GAN based on the BigGAN-PyTorch repository with minor modifications. We do not add extra regularization techniques to stabilize the training. Indeed, ADC-GAN performs more stable during training compared to previous classifier-based cGANs, we argue that the reason is that the classifier of ADC-GAN provides harmonious objective for the generator to learn the data distribution which AC-GAN and TAC-GAN provide contradictory objective (please refer to Table 1).
>
> **Q4**: It would be better if the author could discuss the relationship between ImAC-GAN and ADC-GAN.
>
> **R4**: The output dimension of the ImAC-GAN classifier is C+1 while ours is C*2. ImAC-GAN improves AC-GAN by adding a fake class (say 0) to the classifier Cim: X --> Y U {0}, and formulates the objective functions with the improved classifier as follows:
>
>
>
> max_Cim E_x,y\~p(x,y) [log Cim (y|x)] + E_x\~q(x) [log Cim (0|x)],
>
> max_G E_x,y\~q(x) [log Cim (y|x)].
>
>
>
> Define pm(x,y) = 1/2 p(x,y), for y \in {1,2,...,C} and pm(x,0) = 1/2 q(x), then we can obtain the new optimal classifier
>
> max_Cim E_x,y\~p(x,y) [log Cim (y|x)] + E_x\~q(x) [log Cim (0|x)]
>
> => max_Cim E_x,y\~pm(x,y) [log Cim (y|x)]
>
> => min_Cim E_x\~pm(x) E_y\~pm(y|x) [- log Cim (y|x)]
>
> => min_Cim E_x\~pm(x) [H(pm(y|x)) + KL(pm(y|x) || Cim (y|x))]
>
> => Cim* (y|x) = pm(y|x) => Cim* (y|x) = p(x,y) / (p(x) + q(x)), for y in {1,2,...,C}
>
>
>
> Now, optimizing the generator with the optimal classifier is equivalent to
>
> max_G E_x,y\~q(x,y) [log Cim* (y|x)]
>
> => max_G E_x,y\~q(x,y) [log p(x,y) / (p(x) + q(x))]
>
> => min_G E_x,y\~q(x,y) [log q(x,y) / p(x,y) * (p(x) + q(x)) / q(x,y)] > E_x,y\~q(x,y) [log q(x,y) / p(x,y) * sqrt(p(x) + q(x)) / q(x,y)] + log 2
>
> => min_G E_x,y\~q(x,y) [log q(x,y) / p(x,y) * sqrt(p(x) / q(x)) * q(x) / q(x,y)]
>
> => min_G E_x,y\~q(x,y) [log q(x,y) / p(x,y)] + 1/2 E_x\~q(x) [log p(x) / q(x)] + E_x,y\~q(x,y) [log q(x) / q(x,y)]
>
> => min_G KL(q(x,y)||p(x,y)) - 1/2 KL(q(x)||p(x)) + H(q(y|x))
>
>
>
> In general, the classifier of ImAC-GAN enforces the generator to minimize an upper-bound of KL(q(x,y)||p(x,y)) - 1/2 KL(q(x)||p(x)) + H(q(y|x)) which still contains the drawbacks (i.e., contradictory divergences and unwanted conditional entropy) of AC-GAN as we discussed in Section 2.2.
>
>
>
> **Q5**: In Figure 4 (c) and (f), the results of T-SNE visualization are not very convincing. In my opinion, the author should not use the discriminator to extract feature representations, because the PD-GAN discriminator is less supervised than ADC-GAN. I suggest that the author use a pre-trained classification model to extract features for a fair comparison.
>
> **R5**: We provide T-SNE visualization to investigate whether the discriminator/classifier (they are shared with each other except the output layer) captures the data-to-label relationships. Therefore, we need to use the discriminator/classifier to extract feature representations. Indeed, the PD-GAN discriminator is less supervised than ADC-GAN, and thus can hardly capture the data-to-label relationships, which is a weakness in conditional generative modeling compared with our ADC-GAN.
>
> **Q6**: In the caption of Figure 4, the author says that the T-SNE uses training data, but in the last paragraph of section 5.3, the author says that the T-SNE uses the validation data.
>
> **R6**: The T-SNE visualizes the training set in Figure 4 and we will fix the typo in the updated version.

---

> > ### Comment · Reviewer_uPwH · 2021-11-29
> > **Reply**
> >
> > Thanks for your response. ADCGAN is a stable version of ACGANs (only change the loss function). However, the current paper lacks complete ImageNet experiments. Therefore, I am not sure whether ADCGAN still performs well on high-resolution images of the ImageNet dataset. Anyway, ADCGAN provides a theoretical perspective for us to understand ACGANs, which is the author's core contribution. So I raise my score to 'accept'.

---

### Official Review · Reviewer_DPgR · 2021-11-02

**Correctness:** 3
**Technical Novelty And Significance:** 1
**Empirical Novelty And Significance:** 2
**Recommendation:** 5
**Confidence:** 5

**Main Review:**

Strengths:

(+) The paper exactly points out the primitive problem of ACGAN from the optimization perspective. Since ACGAN is widely adopted in the machine learning area, analyzing problems of ACGAN is necessary and valuable.

(+) The proposed auxiliary discriminative classifier is reasonable, and easy to implement. Also, ADC-GAN does not require much computational burden.

(+) Section 4.2 is very interesting and the explanation of why projection discriminator fails to approximate the joint distribution in Figure 2 is reasonable.

Weaknesses:

(-) It seems that Theorem 1 has already been covered in TAC-GAN paper (paragraphs below eq.4 of the TAC-GAN paper [R1]). Although mathematical formulations are different from each other, the arguments of Theorem 1 and the paragraphs seem to be very similar. I think it is essential to clarify differences between two arguments.

(-) It seems that all experiments were conducted once. It would be better to conduct experiments several times since GANs have been known to have a large performance variance.

(-) The contribution that ADC-GAN can generate diverse images compared to ACGAN and TAC-GAN is not fully demonstrated. Although FID has been a widely used metric to measure fidelity and diversity of generated images, I think It is not enough. I recommend the authors to utilize the improved precision and recall [R2], classification accuracy score [R3], or density and coverage [R4] to quantify the ability of generating diverse images of ADC-GAN.

(-) In section 5.1, the authors conducted the distribution learning experiment using one-dimensional conditional gaussians whose supports are overlapped. I accept that ADC-GAN can learn the joint distribution which consists of the one-dimensional conditional gaussians better than PD-GAN, AC-GAN, and TAC-GAN. However, what about a joint distribution which consists of conditional gaussians with disjoint supports? Can ADC-GAN learn the joint distribution better than other cGANs?

**Summary Of The Paper:**

This paper proposes the Auxiliary Discriminative Classifier GAN (ADC-GAN) to eliminate a contractionary objective and conditional entropy in ACGAN generator training. Specifically, the authors mathematically demonstrate that training ACGAN without a discriminative label classifier causes minimizing an undesirable divergence (KL(q(x)||p(x))) which conflicts with the joint distribution matching (KL(q(x,y)||p(x,y))). Also, they insist that the lack of intra-class diversity of ACGAN results from the absence of generator guidance for training the discriminator. To resolve all these issues, they devise a new classifier, the auxiliary discriminative classifier and deploy the new classifier directly on the ACGAN framework. Experiments demonstrate that ADC-GAN can successfully learn the joint distribution whose conditional marginals have non-negligible support overlap using MoG dataset. In addition, they show the effectiveness of ADCGAN compared to ACGAN, projection discriminator, and TAC-GAN on four benchmark datasets (CIFAR10, CIFAR100, Tiny-ImageNet, and ImageNet) using IS, FID, iFID metrics.

**Summary Of The Review:**

The authors propose a new type of ACGAN named ADC-GAN to address an improper optimization process of ACGAN. They apply adversarial training not only for the discriminator but also for the auxiliary classifier to eliminate a contradictory divergence and conditional entropy in ACGAN training. In the experimental results, they prove the effectiveness of ADC-GAN using synthetic datasets and various benchmark datasets. However, I think Theorem 1 has already been addressed in TAC-GAN paper and experimental results do not fully demonstrate the effectiveness of the proposed method.

---

> ### Author Response · Authors · 2021-11-21
> **Response to Reviewer DPgR**
>
> Thank you for your efforts in reviewing this paper. We respond to your concerns to clarify the theoretical contribution made in this work and highlight the effectiveness of the proposed ADC-GAN.
>
> **Q1**: It seems that Theorem 1 has already been covered in TAC-GAN paper (paragraphs below eq.4 of the TAC-GAN paper [R1]). Although mathematical formulations are different from each other, the arguments of Theorem 1 and the paragraphs seem to be very similar. I think it is essential to clarify differences between two arguments.
>
> **R1**: Our Theorem 1 reveals more than Theorem 1 in TAC-GAN. TAC-GAN's theorem just shows that AC-GAN will induce degenerate conditional distribution $Q_{Y|X}$ even when $Q_X=P_X$. In addition to revealing this issue, our theorem also shows that AC-GAN attempts to maximize $\text{KL}(Q_X\|P_X)$, which is contrary to conditional generative modeling and is not stated in the TAC-GAN paper. The contradictory divergence leads to the training stability and non-robustness with respect to $\lambda$ of TAC-GAN (see Figure 4).
>
> **Q2**: It seems that all experiments were conducted once. It would be better to conduct experiments several times since GANs have been known to have a large performance variance.
>
> **R2**: We ran two new experiments on CIFAR-10 and CIFAR-100 during the rebuttal period. With the previous results, there are now three trails of all methods on CIFAR-10 and CIFAR-100. We here report the means and standard deviationtions over the three training runs. Experiments on Tiny-ImageNet are ongoing and the results will be reported in the updated version.
>
> | FID       | PD-GAN         | AC-GAN          | TAC-GAN         | ADC-GAN                 |
> | --------- | -------------- | --------------- | --------------- | ----------------------- |
> | CIFAR-10  | $6.30\pm 0.19$ | $6.81\pm 0.11$  | $5.88\pm 0.14$  | $\mathbf{5.75}\pm 0.14$ |
> | CIFAR-100 | $8.64\pm 0.20$ | $11.58\pm 0.03$ | $10.99\pm 0.29$ | $\mathbf{7.94}\pm 0.03$ |
>
> **Q3**: The contribution that ADC-GAN can generate diverse images compared to ACGAN and TAC-GAN is not fully demonstrated. Although FID has been a widely used metric to measure fidelity and diversity of generated images, I think It is not enough. I recommend the authors to utilize the improved precision and recall [R2], classification accuracy score [R3], or density and coverage [R4] to quantify the ability of generating diverse images of ADC-GAN.
>
> **R3**: We report the improved precision (P), recall (R), density (D), and coverage (C) results on CIFRA-10 and CIFAR-100 in the following table.
>
> | P, R, D, C | PD-GAN                     | AC-GAN                     | TAC-GAN                    | ADC-GAN                                    |
> | ---------- | -------------------------- | -------------------------- | -------------------------- | ------------------------------------------ |
> | CIFAR-10   | 0.772, 0.646, 0.999, 0.887 | 0.767, 0.618, 0.981, 0.884 | 0.756, 0.647, 0.994, 0.883 | 0.754, **0.686**, 0.958, **0.891**         |
> | CIFAR-100  | 0.743, 0.649, 0.868, 0.826 | 0.730, 0.538, 0.766, 0.754 | 0.740, 0.545, 0.812, 0.775 | **0.772**, **0.652**, **0.949**, **0.845** |
>
> The proposed ADC-GAN achieves the best recall (R) and coverage (C) results, verifying that it improves the diversity of generated samples. ADC-GAN also obtains the best precision (P) and density (D) scores on CIFAR-100, showing better image fidelity on the more fine-grained dataset.
>
> **Q4**: In section 5.1, the authors conducted the distribution learning experiment using one-dimensional conditional gaussians whose supports are overlapped. I accept that ADC-GAN can learn the joint distribution which consists of the one-dimensional conditional gaussians better than PD-GAN, AC-GAN, and TAC-GAN. However, what about a joint distribution which consists of conditional gaussians with disjoint supports? Can ADC-GAN learn the joint distribution better than other cGANs?
>
> **R4**: To evaluate the performance on MoG with disjoint supports, we set the MoG dataset with $\mu_0=0, \mu_1=6, \mu_2=12$ and $\sigma_0=1,\sigma_1=1,\sigma_2=1$. The following table reports the MMD results of different methods with and without the GAN loss.
>
> | Log Loss | PD-GAN | AC-GAN | TAC-GAN | ADC-GAN |
> | -------- | ------ | ------ | ------- | ------- |
> | Class_0  | 0.0048 | 0.0432 | 0.0142  | 0.0255  |
> | Class_1  | 0.9581 | 0.4196 | 0.1940  | 0.0020  |
> | Class_2  | 3.5979 | 0.3317 | 0.6559  | 0.7495  |
> | Marginal | 0.7329 | 0.0848 | 1.3020  | 0.0621  |
>
> In general, our ADC-GAN can faithfully learn the real joint distribution regardless of its shape.

---

> > ### Comment · Reviewer_DPgR · 2021-11-27
> > **Reply**
> >
> > Thanks for your response which addresses my concerns. I will be grateful if you add the experiments and discussions done during the rebuttal period in your final paper. However, there are still remaining concerns regarding inconsistent numbers from the results of PyTorch-StudioGAN library. Also, I have understated that the auxiliary discriminative classifier can be applied to any type of ACGAN variants, such as ContraGAN [1], OmniGAN [2], and ReACGAN [3]. I think conducting more experiments using those GANs will make ADCGAN more convincing. Anyway, I'm going to raise my score by one point.
> >
> > [1] Kang, M., & Park, J. (2020). ContraGAN: Contrastive Learning for Conditional Image Generation. Neurips.
> >
> > [2] Zhou, P., Xie, L., Ni, B., Geng, C., & Tian, Q. (2021). Omni-gan: On the secrets of cgans and beyond. CVPR.
> >
> > [3] Kang, M., Shim, W., Cho, M., & Park, J. (2021). Rebooting ACGAN: Auxiliary Classifier GANs with Stable Training. Neurips.

---

> > > ### Comment · Reviewer_DPgR · 2021-12-05
> > > **I have a major concern about this paper**
> > >
> > > I found that Eq. (5) in this paper is very similar to Eq. (8) in the [paper](https://arxiv.org/abs/2106.08601v4) "self-supervised gans with label augmentation", which was published on arXiv on 2021.06.16. Since the archive publication date is before the ICLR submission deadline, I think the authors should have clarified the differences between these two papers.
> > > Both studies (ADCGAN and SSGAN-LA) tackle the generator-agnostic optimization process of GANs and insist that the process make the generator's implicit distribution converge to a degenerate distribution. To remove the problem, both works propose discriminative classifiers, which are trained using Eq. (5) and Eq. (8).
> > >
> > > For these reasons, I think the technical contributions of this paper are not significant, and the authors do not adequately describe the closely related work. I rearrange my score from 6 to 5.

---

> > > > ### Author Response · Authors · 2021-12-06
> > > > **Authors' Response**
> > > >
> > > > ADC-GAN and SSGAN-LA solve problems in different fields respectively, although their solutions are similar in mathematical formulation. SSGAN-LA belongs to self-supervised GANs that improve the training stability of unsupervised GANs by introducing self-supervised tasks. ADC-GAN focuses on solving the low intra-class diversity issue of AC-GAN, which is an important issue in the class-conditional GAN literature.
> > > >
> > > > In addition to proposing ADC-GAN, the contribution of this paper also includes analyzing the drawbacks of previous conditional GANs (e.g., AC-GAN, TAC-GAN, PD-GAN) as well as the advantages of ADC-GAN. Besides, the analysis results and proof are different from that in SSGAN-LA.
> > > >
> > > > We will clarify the difference between ADC-GAN and SSGAN-LA clearly in the updated version. In general, ADC-GAN provides a theoretically grounded and superior (v.s prior work) solution to implement conditional GANs, and we argue that ADC-GAN will have a broad and significant impact on this field.

---

> > > > > ### Comment · Reviewer_DPgR · 2021-12-07
> > > > > **Reply**
> > > > >
> > > > > ADC-GAN and SSGAN-LA are devised to solve problems in different fields (conditional GAN and self-supervised GANs), but these two fields can be grouped under the theme of GAN. Also, both models address the same problem, specifically the generator-agnostic optimization process of GAN. So, I stick with my previous score as the technical contributions of this paper are not sufficient.

---

### Official Review · Reviewer_ebJs · 2021-11-02

**Correctness:** 3
**Technical Novelty And Significance:** 2
**Empirical Novelty And Significance:** 2
**Recommendation:** 6
**Confidence:** 4

**Main Review:**

## Strengths
1. The paper presents interesting analysis of AC-GAN, TAC-GAN, PD-GAN. Especially the Theorem 3 reveals potential drawbacks of TAC-GAN.
2. The experimental results on synthetic and real datasets demonstrate the superiority of ADC-GAN on conditional generative modeling tasks.

## Weaknesses
1. I don't fully agree with some claims made by the authors:
   1. Page 5, footnote 2, it might be true that term (a) is "ignored" or set to zero, but it is not sufficient to say this inductive bias is a mistake.
2. Equation 8, this is not the original form of TAC-GAN. The original TAC-GAN is built upon AC-GAN, so term (c) in Equation 3 in the TAC-GAN paper is missing. I notice that Equation 8 is consistent with the actual implementation of TAC-GAN, but I guess it is good to state this clearly in the paper.
3. Implementations of ADC-GAN and TAC-GAN are the same? As I checked the provided code in supplementary, I think the proposed ADC-GAN is very similar to TAC-GAN (as defined in Equation 8): In fact, if spectral norm (SN) and bias are not used in the linear classification layer, they are exactly equivalent. This is because the weight of $C_d$ is just $C$ and $C_{mi}$ stacked together. I would consider this as an implementation difference. Note that Theorem 2 and 3 are different, I doubt the superior performance of ADC-GAN might come from the difference in SN or a different choice of hyperparameters. In such case, it would be helpful if the author could provide code for MoG experiment, which is cleaner, simpler, and no SN applied (if the code is borrowed from TAC-GAN). Please correct me if I am wrong, and I'm happy to amend my score accordingly.
4. It would be helpful if the author could provide results of ADC-GAN on ImageNet at 256 resolution.
5. Table 2, why not use the reported numbers in TAC-GAN paper? I checked Table 1 in TAC-GAN paper, and their FID on CIFAR100 is 7.22 which is lower than the reported 7.98, any explanation?
6. It is nice to see ADC-GAN worked even without GAN loss, I am curious how the model performs (w/o GAN loss) on challenging datasets such as ImageNet? It is also surprising that in Supplementary Table 4, Hinge loss does worse than no GAN loss: if Theorem 2 holds, the solution set of ADC-GAN is a subset of (unconditional) GAN (say with hinge loss), so adding GAN loss wouldn't affect ADC-GAN training. Can the author explain this?
7. Comparing Theorem 2 with a plain cGAN which minimizes $JS(Q_{X,Y}||P_{X,Y})$, does the reverse KL tend to cause mode collapse? (in theory it seems that JS is better than reverse KL? a typical yet imprecise point to be made is that KL causes mode averaging and reverse KL causes mode collapse. [1])

[1] Zhao, Miaoyun, et al. "Bridging Maximum Likelihood and Adversarial Learning via α-Divergence." Proceedings of the AAAI Conference on Artificial Intelligence. Vol. 34. No. 04. 2020.

**Summary Of The Paper:**

This paper proposes a new conditional GAN model that employs a discriminative classifier that predicts in the joint space of label and real/fake domain. The theoretical analysis shows the proposed ADC-GAN can minimize the reverse KL between joint $Q_{X,Y}$ and $P_{X,Y}$.

**Summary Of The Review:**

I think the paper presents an interesting analysis of TAC-GAN, AC-GAN, and other cGAN methods. My concern is that the proposed method has the same actual implementation as existing method (TAC-GAN). The paper also lacks results on high-resolution generation. I am willing to raise my score if my concerns are resolved.

---

> ### Author Response · Authors · 2021-11-21
> **Response to Reviewer ebJs**
>
> Thank you for spending time reviewing our paper and your insightful comments. We sincerely hope that our clarification will resolve your concerns.
>
> **Q1**: It might be true that term (a) is "ignored" or set to zero, but it is not sufficient to say this inductive bias is a mistake.
>
> **R1**: We will refine our claim into that the term (a) is ignored in PD-GAN's implementation. This inductive bias may not be a mistake but it will adversely affect the results as shown in P2GAN[A] that optimizing term (a) additionally would improve the performance.
>
> **Q2**: Equation 8 is consistent with the actual implementation of TAC-GAN. It is good to state this clearly in the paper.
>
> **R2**: We would like to thank you for your valuable suggestion. Indeed, our Equation 8 corresponds to the actual implementation of TAC-GAN. We will state it clearly in the updated version.
>
> **Q3**: **Implementations of ADC-GAN and TAC-GAN are the same?** Can we construct C_d of ADC-GAN by stacking C and C_mi of TAC-GAN?
>
> **R3**: ADC-GAN and TAC-GAN are indeed different. Recall that the two classifiers of TAC-GAN at their optimum can be written in the form of
>
> $$C^*(y|x)=\frac{p(x,y)}{p(x)}=\frac{\exp(v_y^p\cdot \phi(x))}{\sum_{k} \exp(v_{k}^p\cdot \phi(x))},C_{mi}^*(y|x)=\frac{p(x,y)}{p(x)}=\frac{\exp(v_y^q\cdot \phi(x))}{\sum_{k} \exp(v_{k}^q\cdot \phi(x))},$$
>
> which conclude
>
> $$\exp(v_y^p\cdot\phi(x))=k_1 p(x,y),\sum_k\exp(v_k^p\cdot\phi(x))=k_1p(x),$$
>
> $$\exp(v_y^q\cdot\phi(x))=k_2 q(x,y),\sum_k\exp(v_k^q\cdot\phi(x))=k_2 q(x),$$
>
> for $k_1,k_2\in\mathbb{R}^+$. If we stack $C$ and $C_{mi}$ together, then the stacked classifier will be
>
> $$C_s^*(y,1|x)=\frac{\exp(v_y^p\cdot \phi(x))}{\sum_{k} \exp(v_{k}^p\cdot \phi(x))+\sum_{k} \exp(v_{k}^q\cdot \phi(x))}=\frac{k_1 p(x,y)}{k_1 p(x)+k_2 q(x)},$$
>
> $$C_s^*(y,0|x)=\frac{\exp(v_y^q\cdot \phi(x))}{\sum_{k} \exp(v_{k}^p\cdot \phi(x))+\sum_{k} \exp(v_{k}^q\cdot \phi(x))}=\frac{k_2 q(x,y)}{k_1 p(x)+k_2 q(x)},$$
>
> which is different from the optimal discriminative classifier of ADC-GAN $C_d^*(y,1|x)=\frac{p(x,y)}{p(x)+q(x)},C_d^*(y,0|x)=\frac{q(x,y)}{p(x)+q(x)}$ if $k_1\neq k_2$, which is quite possible (the probability is almost 100%). Therefore, it is almost impossible to stack C and Cmi to construct Cd.
>
> **Q4**: It would be helpful if the author could provide results of ADC-GAN on ImageNet at 256 resolution.
>
> **R4**: Due to our limited computational resources, we are sorry that we cannot provide results of ADC-GAN on ImageNet at 256 resolution during the short rebuttal period.
>
> **Q5**: Table 2, why not use the reported numbers in TAC-GAN paper?
>
> **R5**: We implement all methods in Table 2 based on the official BigGAN-PyTorch repository. The experimental settings are different from TAC-GAN. For example, the batch size is 50 instead of 100 used in TAC-GAN, and the discriminator update steps are 4 per generator step while it is 2 in TAC-GAN. And we use a single GPU to conduct these experiments due to our limited computational resources while TAC-GAN uses two GPUs. Besides, we cannot reproduce the results of TAC-GAN on one GPU using their code. In general, our comparison is fair as we implement all methods using the same experimental settings.
>
> **Q6**: I am curious how the model performs (w/o GAN loss) on challenging datasets such as ImageNet? It is also surprising that in Supplementary Table 4, Hinge loss does worse than no GAN loss. Can the author explain this?
>
> **R6**: Indeed, Figure 4(a,d) shows that ADC-GAN w/o GAN loss on CIFAR-10 and CIFAR-100 can achieve comparable performance with the full ADC-GAN model. However, due to our limited computational resources, conducting experiments on ImageNet during the short rebuttal period is difficult for us. As for the results in Table 4, we suspect that the reason is that the MoG dataset is too simple for hinge loss to have an advantage over others. Nevertheless, our ADC-GAN can outperform others when all methods use hinge loss.
>
> **Q7**: Comparing Theorem 2 with a plain cGAN which minimizes JS(QX,Y||PX,Y), does the reverse KL tend to cause mode collapse?
>
> **R7**: In our experiments, our method is more stable than PD-GAN during training and hardly encounters mode collapse (shown in Figure 4). Our Theorem 2 aims to show that ADC-GAN has the unique global optimal solution ($Q_{X,Y}=P_{X,Y}$). Note that reverse KL tends to cause mode collapse only when the model capability is not enough, which can be easily resolved by enlarging the model size. In addition, our ADC-GAN contains the original discriminator that helps to minimize $\text{JS}(P_X | |Q_X)$. which is symmetric that can prevent the generator from mode collapse.
>
> [A] Han, Ligong, et al. "Dual Projection Generative Adversarial Networks for Conditional Image Generation." *Proceedings of the IEEE/CVF International Conference on Computer Vision*. 2021.

---

> > ### Comment · Reviewer_ebJs · 2021-11-27
> > **Further comments**
> >
> > I thank the authors for their clarification and explanations. I understand that given the period of time and limited resources it can be hard to run additional experiments. Since my main concern about ADC-GAN and TAC-GAN being equivalent is resolved, I'm raising my score to 6, but I'm not willing to champion it.
> >
> > In my previous comment, I ignored the fact that in a "stacked" classifier, logits of the fake-classifier are also being normalized in the softmax when computing the real-classificaiton loss (the same goes for the fake-classification loss). I guess the performance gain comes from this different normalization in the denominator. I think this might be a valid trick, although I am still not convinced by its theoretical analysis. I highly suggest the authors do some more analysis in this direction (without training on more datasets or higher resolutions). An example would be visualizing feature norm and grad norm as done in ReACGAN (Figure 2). Also, it would be good if the authors provided more results/analysis on ADC-GAN without unconditional GAN loss.

---

### Official Review · Reviewer_mZT7 · 2021-11-02

**Correctness:** 3
**Technical Novelty And Significance:** 2
**Empirical Novelty And Significance:** 2
**Recommendation:** 5
**Confidence:** 5

**Main Review:**

Strength:

* The proposed discriminative classifiers seems interesting to resolve the biased issue of ACGAN.

Weakness:

* The key results seems to be Thm 2, which is based on Prop. 2. However, the proof shows pm(x, y, l) = pm(x, y, 1) + pm(x, y, 0) = 1/2 p(x, y) + 1/2q(x, y). at the very beginning. How do you get the second equation?  I guess it may not be a fatal error, but I can't tell the correctness at the moment.

* The main criticism of AC-GAN is being generator-agnostic, which I think it's not fully appropriate. A common practical implementation of AC-GAN is also using generated data to train classifier, which I think it's a straightforward idea. Under this, whether it can simply resolve the generator-agnostic issue? However, by just doing so, the performance seems not as competitive as the reported numbers of the proposed method.  Could you comment on it?

* The reported numbers looks not quite consistent with other works to me.

  - For CIFAR10 results, the reported FID are all below 7, however, it's not the case for most of the existing works. For example, all the models reported here https://github.com/POSTECH-CVLab/PyTorch-StudioGAN#cifar10-3x32x32 are all with FID > 7.

 - For CIFAR100, the numbers reported in TAC-GAN seems better than the 11.37 reported in the paper for the TAC-GAN and 7.98 for the proposed method.

- For ImageNet, again the reported BigGAN is worse than it should be.  For example, see Table 2 in https://arxiv.org/pdf/2111.01118v1.pdf which provides a nice comparison.

* Some descriptions are not accurate. For example, there are multiple sentences about "GANs are notoriously unstable to train", which is true back to 2014. However, there are already "tons" of papers working on it to resolve the issues. The authors should at least cite those and make a fair description.  Secondly, the below Eq (6), the authors mention they proposed method can be "unbiasedly optimize", which is not true under alternative and minibatch setting. Again, there are lots of works (check all the works on GAN optimization) discussing this issue. The authors should remove this incorrect claim.

* Proposition 1 seems trivial. Also, Theorem 1 is quite similar to the analysis in TAC-GAN. I would highly suggest removing Proposition 1,  which you don't have to pretend to be a theoretical paper. Also, cite TAC-GAN for the analysis. We should not copy or redo the analysis from the predecessor.

* I have the concern of the analysis. Most of the results rely on assuming sth components are optimal. However, in reality, they are not hold in reality, and there is no convergence analysis provided.  Could the authors comment on it?

* The biased issue of ACGAN is known.  For example,

  - AC-GAN Learns a Biased Distribution, 2017
 - Unbiased Auxiliary Classifier GANs with MINE, 2020

 There are many more. The authors should provide a better overview for the progress of this direction.


Question:

* In Table 1, PD-GAN is optimizing JS(P||Q) while the proposed ADC-GAN is optimizing KL(P||Q).  To me, there should not much difference. Any insights why the proposed ADC-GAN, which optimizes an asymmeric loss, should be better?

Suggestion:

* There are some very recent works in NeurIPS and also highly relayed,
  - Rebooting ACGAN: Auxiliary Classifier GANs with Stable Training, NeurIPS 2021
  - A Unified View of cGANs with and without Classifiers, NeurIPS 2021
  Although they are posted online after ICLR deadline, I would strongly encourage the authors comment on the similarities and differences between the proposed work and these two, because they will be on public for a when the paper decision of ICLR is out anyway. Note that I won't judge the paper decision based on this, but I think it's great to have for the community. It's fine even though the ideas are overlapping.

**Summary Of The Paper:**

The paper is about improving conditional GANs. To be specifically, it also aims to resolve the bias issue of ACGAN by proposing a discriminative classifier. The discriminative classifier is a hybrid model of discriminator and classifier, where it has to not tell real or fake, but also the class.  Preliminary analysis of the proposed method are provided. Experiments are conducted on the standard benchmarks.

**Summary Of The Review:**

The main concerns are first on the analysis, which I couldn't tell the correctness at the moment. Second issue is the empirical results, which seems not consistent with other works.

---

> ### Author Response · Authors · 2021-11-21
> **Response to Reviewer mZT7 [1/2]**
>
> We would like to thank the reviewer for spending time reviewing our paper and providing constructive feedback. We here respond to each weakness raised by the reviewer.
>
> **Q1**: The key results seems to be Thm 2, which is based on Prop. 2. However, the proof shows pm(x, y, l) = pm(x, y, 1) + pm(x, y, 0) = 1/2 p(x, y) + 1/2q(x, y). at the very beginning. How do you get the second equation?
>
> **R1**: The second equation comes from our definition of pm. We define pm(x,y,1) = 1/2 p(x,y) and pm(x,y,0) = 1/2 q(x,y). We will clarify it more clearly in the updated version.
>
> **Q2**: The main criticism of AC-GAN is being generator-agnostic, which I think it's not fully appropriate. A common practical implementation of AC-GAN is also using generated data to train classifier. Under this, whether it can simply resolve the generator-agnostic issue? Could you comment on it?
>
> **R2**: As we discussed in Appendix B that the original AC-GAN could also suffer from the generator-agnostic issue of our used "stable" AC-GAN. As the authors of TAC-GAN said in https://github.com/batmanlab/twin-auxiliary-classifiers-gan/issues/1, training the auxiliary classifier with fake data may influence the classification accuracy of the auxiliary classifier due to that the fake data is poor quality at the beginning, and thus there is no advantage to training the auxiliary classifier on the fake data. In addition, the authors of ReACGAN also suggested in https://github.com/POSTECH-CVLab/PyTorch-StudioGAN/issues/74 that removing conditional loss on fake images when training the discriminator (classifier) will enable ACGAN to generate more realistic images. Many other papers also use the "stable" version of AC-GAN (e.g., Eq. 6,7,8,9 in https://arxiv.org/pdf/1703.02000.pdf, Eq. 2,3,4 in https://arxiv.org/pdf/1811.11163.pdf, Eq. 3,4,5,6 in https://arxiv.org/pdf/2105.05501.pdf). In summary, training the classifier of AC-GAN without the generated data is also a common practice and can yield better performance.
>
> **Q3**: The reported numbers looks not quite consistent with other works to me.
>
> **R3**:
>
> - For CIFAR-10, we implement methods and conduct experiments on CIFAR-10 based on the official BigGAN-PyTorch repository, which is different from the PyTorch-StudioGAN repository in aspects of training settings and evaluation protocols.
>
> - For CIFAR-100, we implement methods and conduct experiments on CIFAR-100 based on the official BigGAN-PyTorch repository, which is slightly different from the TAC-GAN codebase. For example, the batch size is 50 instead of 100 used in TAC-GAN, and the discriminator update steps are 4 per generator step while it is 2 in TAC-GAN. And we use a single GPU to conduct these experiments due to our limited computational resources while TAC-GAN uses two GPUs. Besides, we cannot reproduce the results of TAC-GAN on one GPU using their code.
>
> - For ImageNet, the results of BigGAN/PD-GAN and TAC-GAN on ImageNet are from the TAC-GAN paper, where the experimental settings such as the number of channels and training iterations are different from the original BigGAN. We follow the experimental settings of TAC-GAN (due to our limited computational resources) to run ADC-GAN on ImageNet.
>
> In summary, our comparison is fair as we implement all methods using the same experimental settings.
>
>
>
> **Q4**: Some descriptions are not accurate. For example, there are multiple sentences about "GANs are notoriously unstable to train", which is true back to 2014. However, there are already "tons" of papers working on it to resolve the issues. The authors should at least cite those and make a fair description. Secondly, the below Eq (6), the authors mention they proposed method can be "unbiasedly optimize", which is not true under alternative and minibatch setting. Again, there are lots of works (check all the works on GAN optimization) discussing this issue. The authors should remove this incorrect claim.
>
> **R4**: We would like to thank the reviewer for valuable feedback and will refine these claims in the updated version.
>
> **Q5** Proposition 1 seems trivial. Also, Theorem 1 is quite similar to the analysis in TAC-GAN.
>
> **R5**: Our Theorem 1 reveals more than Theorem 1 in TAC-GAN. TAC-GAN's theorem just shows that AC-GAN will induce degenerate conditional distribution q(y|x) even when q(x)=p(x). In addition to revealing this issue, our theorem also shows that AC-GAN attempts to minimize -KL(q(x)||p(x)), which is contrary to conditional generative modeling and is not stated in the TAC-GAN paper. The contradictory divergence leads to the training stability and non-robustness with respect to $\lambda$ of TAC-GAN (see Figure 4).

---

> > ### Comment · Reviewer_mZT7 · 2021-11-29
> > **Thanks for the response. I decide to keep my score.**
> >
> > Thanks for your response.  Some of my concerns are addressed. However, as ICLR allows authors to update the draft, and I don't see any updates in the draft has been made. Also, for the experiment, as GAN are sensitive to the hyperparameters, it's not convincing to me that you use your own setting instead of the standard protocols given you are able to train on ImageNet.

---

> > > ### Author Response · Authors · 2021-11-30
> > > **Thank you and further response**
> > >
> > > Thank you for your reply. First, we apologize for ignoring the fact that ICLR allows us to update drafts directly. We will revise our paper according to our responses in the updated version. As for experiments on CIFAR-10/100, we follow the experimental settings of BigGAN-PyTorch, which is also a standard protocol in the GAN literature. As for experiment on ImageNet, we adopted the settings of TAC-GAN to make the comparison as fair as possible and due to our limited and temporarily available computational resources (TAC-GAN modifies $ch=96\rightarrow 64$). TAC-GAN is an appropriate competitor to our ADC-GAN because it also belongs to the classifier-based cGANs. And ADC-GAN surpasses TAC-GAN, as well as PD-GAN, under the same setting.

---

> ### Author Response · Authors · 2021-11-21
> **Response to Reviewer mZT7 [2/2]**
>
> **Q6**: I have the concern of the analysis. Most of the results rely on assuming sth components are optimal. However, in reality, they are not hold in reality, and there is no convergence analysis provided. Could the authors comment on it?
>
> **R6**: We follow the common practice in the GAN literature (e.g., the original GAN paper) to analyze the intrinsic learning objective for the generator based on the optimal discriminator/classifier. We leave the convergence analysis as the future work as this is not a fully theoretical paper.
>
> **Q7**: The biased issue of ACGAN is known. For example,
>   - AC-GAN Learns a Biased Distribution, 2017
>   - Unbiased Auxiliary Classifier GANs with MINE, 2020
>
> **R7**: In this paper, we give a new explanation of the biased issue of AC-GAN that the classifier of AC-GAN is generator-agnostic (and thus cannot provide only the discrepancy between q(x,y) and p(x,y) for the generator). And we also show in Theorem 1 that AC-GAN optimizes contradictory divergence, which is not stated in previous work. However, we would like to thank the reviewer for constructive comments and will review the related work in more detail and systematically in the updated version.
>
> **Q8**: Why the proposed ADC-GAN (KL(P||Q)) is better than PD-GAN (KL(P||Q))?
>
> **R8**: As we discussed in Section 4.2 and shown in Figure 4(c,f), PD-GAN is incapable of modeling the data-to-label relationships since it ignores the partition term (a). We argue that modeling the data-to-label relationships is beneficial for conditional generative modeling. And ADC-GAN can therefore achieve better results due to modeling such relationships. In addition, ADC-GAN also minimizes JS(p(x)||q(x)), which is also symmetric, in addition to the KL(p(x,y)||q(x,y)).
>
> **Q9**: There are some very recent works in NeurIPS and also highly relayed,
>
>   - [1] Rebooting ACGAN: Auxiliary Classifier GANs with Stable Training, NeurIPS 2021
>   - [2] A Unified View of cGANs with and without Classifiers, NeurIPS 2021
>
> **R9**:
>
> ReACGAN [1] essentially does not solve the problem of low intra-class diversity in AC-GAN because it still inherits the non-discriminative classifier of AC-GAN. The ReACGAN authors also acknowledged in their paper (page 19 and 20) that ReACGAN fails to estimate the 1-D MoG dataset, which our ADC-GAN can faithfully replicate.
>
> ECGAN [2] proposed a framework that unifies existing projection-based cGANs (PD-GAN) and classifier-based cGANs (AC-GAN). However, our proposed ADC-GAN does not follow this framework due to the fact that our discriminative classifier is different from the classifier in existing cGANs.
>
> We would like to thank the reviewer for the constructive suggestion and agree that discussing the similarities and differences between the proposed method and more related work is valuable for this paper and the community. We will add the discussion with the recent related work in the updated version.

---

### Decision · Program_Chairs · 2022-01-20

**Decision:**

Reject

**Comment:**

The paper proposes a conditional generative adversarial network with an auxiliary discriminative classifier for conditional generative modeling. The auxiliary discriminative classifier can provide the discrepancy between the joint distribution of the real data and labels and that of the generated data and labels to the generator by discriminatively predicting the label of the real and generated data. Experiment results are provided to demonstrate the effectiveness of the proposed idea.  The current paper receives mixed ratings after rebuttal (5, 6, 5, 8). Except that one reviewer (the Reviewer uPwH) will champion the paper with a score of 8, the concerns of the other three reviewers remain. To be specific, even though Reviewer ebJs assigns a score of 6, he/she doesn’t champion the paper because additional experiments requested are not provided by the authors, including (i) training on more datasets or higher resolutions, (ii) visualizing feature norm and grad norm as done in ReACGAN, (iii) experiments on ADC-GAN without unconditional GAN loss. The Reviewer DPgR pointed out that the paper might have a novelty issue because it bears some similarities with other works but it lacks a discussion in the revision. Additionally, Reviewer mZT7 pointed out that the authors didn’t provide a revised paper during the rebuttal, thus leading to a difficulty to assess the quality of the final paper. As a result, AC thinks that the paper is not ready to publish at the current stage and recommends a rejection.  The AC urges the authors to revise their paper according to the comments provided by the reviewers, and resubmit their work in a future venue.